# *Metarhizium* spp. isolates effective against Queensland fruit fly juvenile life stages in soil

**Madita Prince** [iD] [1]*, **Aimee C. McKinnon**[2], **Diana Leemon**[3], **Tim Sawbridge**[2,4], **John Paul Cunningham**[2,4]

1 Agriculture Victoria, Tatura SmartFarm, Tatura, VIC, Australia, 2 Agriculture Victoria, Agribio Centre for AgriBiosciences, Bundoora, VIC, Australia, 3 Myco-Vation, Bellbowrie, QLD, Australia, 4 School of Applied Systems Biology, La Trobe University, Melbourne, VIC, Australia

* madita.prince@ecodev.vic.gov.au, maditaprince_@hotmail.com

## Abstract

Queensland fruit fly, *Bactrocera tryoni*, Froggatt (Diptera: Tephritidae) is Australia's primary fruit fly pest species. Integrated Pest Management (IPM) has been adopted to sustainably manage this polyphagous species with a reduced reliance on chemical pesticides. At present, control measures are aimed at the adult stages of the fly, with no IPM tools available to target larvae once they exit the fruit and pupate in the soil. The use of entomopathogenic fungi may provide a biologically-based control method for these soil-dwelling life stages. The effectiveness of fungal isolates of *Metarhizium* and *Beauveria* species were screened under laboratory conditions against Queensland fruit fly. In bioassays, 16 isolates were screened for pathogenicity following exposure of third-instar larvae to inoculum-treated vermiculite used as a pupation substrate. The best performing *Metarhizium* sp. isolate achieved an average percentage mortality of 93%, whereas the best performing *Beauveria* isolate was less efficient, with an average mortality of 36%. Susceptibility to infection during different development stages was investigated using selected fungal isolates, with the aim of assessing all soil-dwelling life stages from third-instar larvae to final pupal stages and emerging adults. Overall, the third larval instar was the most susceptible stage, with average mortalities between 51–98% depending on the isolate tested. Moreover, adult mortality was significantly higher when exposed to inoculum during pupal eclosion, with mortalities between 56–76% observed within the first nine days post-emergence. The effect of temperature and inoculum concentration on insect mortality were assessed independently with candidate isolates to determine the optimum temperature range for fungal biological control activity and the rate required for application in field conditions. *Metarhizium spp.* are highly efficacious at killing Queensland fruit fly and have potential for use as biopesticides to target soil-dwelling and other life stages of *B. tryoni*.

## Introduction

Tephritid fruit flies are becoming increasingly problematic for Australia's $15 billion horticultural industry [1–3]. Queensland fruit fly, *Bactrocera tryoni* Froggatt (Diptera: Tephritidae), is

---

**Data Availability Statement:** All relevant data are within the manuscript and its Supporting Information files.

**Funding:** This research was conducted as part of a PhD scholarship with the School of Applied Systems Biology, La Trobe University (https://www.latrobe.edu.au). The scholarship was funded through the project: A national biocontrol program to manage pest fruit flies in Australia (Project ID: 4-EKSH327), with co-investment from Agriculture Victoria Research and the Department of Agriculture, Fisheries and Forestry. The funders had no role in study design, data collection and analysis, decision to publish, or preparation of the manuscript.

**Competing interests:** The authors have declared that no competing interests exist.

the primary tephritid pest; a highly polyphagous pest that is native to the country and causes considerable losses to production and market access [2, 4]. Queensland fruit fly is endemic to tropical and subtropical areas, but is now established widely throughout eastern Australia, having extended its geographical distribution into more temperate regions of southern Australia [3]. In Victoria, changes in climate over the past decade may be providing habitats that are more suitable for Queensland fruit fly populations, and the local climate is projected to become increasingly suited over time [1, 5].

The last decade seen the restricted use and withdrawal of several chemical insecticides that were previously relied upon for controlling fruit fly pests in Australia [6, 7]. Widespread pesticide use has been replaced by sustainable tools and practices that include protein bait spraying, male annihilation technique (MAT), female mass trapping, and the potential for sterile insect technique (SIT) [8–10]. These strategies target adult flies in the population, and flies that evade these control techniques may continue to reproduce, enabling local populations to persist. A more holistic strategy that targets all life stages of the fly may be more effective [11, 12]. The use of parasitoid wasp species that specifically target the egg and early larval stages of Queensland fruit fly developing in the fruit has been explored as augmentative (mass release) and conservation biological control (habitat improvement) approaches [13, 14]. Good orchard hygiene practices (i.e., removing fallen fruit) can also help reduce late-stage larvae that have escaped parasitism, but once they exit the fruit to pupate in soil, they can complete their development.

Entomopathogenic fungi are microorganisms that can be applied as biological control agents in liquid solutions, as contact powder, or as a bait formulation to infect and kill arthropod pests [15, 16]. Two entomopathogenic fungal species of the order Hypocreales, *Metarhizium anisopliae* complex (Metschn.) Sorokin (family: Clavicipitaceae) and *Beauveria bassiana* (Bals.-Criv.) Vuill. (family: Cordycipitaceae) have had the most attention as potential biocontrol agents for tephritid fruit fly control [17]. These entomopathogens are known for their safety [18, 19], ease of mass production [20], and pathogenicity against diverse insect groups, including tephritid fruit flies [21–25]. *M. anisopliae* and *B. bassiana* have been widely researched to manage South American fruit fly (*Anastrepha fraterculus*) [26], Mexican fruit fly (*A. ludens*) [26–29], Mediterranean fruit fly (*Ceratitis capitata*) [23, 30–33], mango fruit fly (*C. cosyra*) [31, 32], olive fruit fly (*Bactrocera oleae*) [34], Oriental fruit fly (*B. dorsalis*) [35, 36], carambola fruit fly (*B. carambolae*) [37, 38], and peach fruit fly (*B. zonata*) [35, 36, 39, 40]. These two entomopathogenic fungi species have been shown to exert pathogenicity against all life stages (larvae, pupae, and adult life stages) of Mediterranean fruit fly (*C. capitata*) under laboratory conditions [23, 30, 33]. Exposure of pupating larvae of Oriental fruit fly (*B. dorsalis*) and olive fruit fly (*B. oleae*) to *M. anisopliae*- or *M. brunneum*-inoculated soil resulted in reduced adult emergence of both flies [11, 34]. When the adult and juvenile life stages of Oriental fruit fly (*B. dorsalis*) and peach fruit fly (*B. zonata*) were tested, the same fungal isolates were highly effective as well [35, 40]. There have been no studies to date reporting on the effectiveness of entomopathogenic fungi against juvenile life stages of Queensland fruit fly, and only one study has reported on the efficacy of a single isolate of *M. anisopliae* against adult Queensland fruit fly [24].

We explored the potential for entomopathogenic fungi to be used as a soil application to control the juvenile life stages of Queensland fruit fly. Focusing on *Beauveria bassiana* and *Metarhizium anisopliae sensu lato* (*M. anisopliae* s.l.) species, fungal isolates were obtained from soil collected from the Australian states of Queensland (eastern) and Victoria (southern). Isolates were screened in bioassays for their pathogenicity against third-instar larvae, pupae, and emerging adults. We developed a bioassay to screen fungal entomopathogenic species and isolates, assessing host mortality rates under laboratory conditions [41]. To improve the

chance of future field efficiency of the prospective isolates, we screened the suitability of isolates through testing the effect of temperature on the isolate growth attributes [16, 42, 43], in addition to their pathogenicity against Queensland fruit fly. The virulence of an isolate likely differs from more sterile conditions (vermiculite) through interactions between the inoculum and resident soil microbiota [44, 45]. To explore this further, the most pathogenic isolate tested was evaluated in non-sterile orchard soil against third-instar larvae.

## Methodology

### Insects

A laboratory culture of Queensland fruit fly was maintained at the Tatura SmartFarm, Victoria, Australia, at 25˚C, 60% relative humidity (RH), and a 16:8 (L:D) photoperiod. Adult flies were housed in fine mesh insect cages (30 cm x 30 cm x 30 cm) where they were provided with water and a mixture of sugar and hydrolysed enzymatic yeast (3:1) as a food substrate [46]. A perforated plastic cup (30 mL with 14 holes 1.1 mm in diameter) lined with apple juice-soaked filter paper (no. 1) was placed for four hours in the cage for mated female flies to oviposit into. Eggs laid on the filter paper were placed onto an artificial carrot-based larval diet (dehydrated carrot (20%), brewer's yeast (6.7%), and methyl 4-hydroxybenzoate (0.67%) per litre sdH$_2$O) in containers (500 mL), with approximately 30 eggs per 15 g of diet. The containers were placed in a second container (7.5 l) filled with 2 cm of vermiculite and secured with a lid. Larvae were left to feed on the diet until they reached the late third-instar "popping" stage, at which point they left the diet to pupate in the provided vermiculite.

### Isolation and identification of entomopathogenic fungi

Fungal entomopathogens were isolated from soil collected from a biodynamic orchard located in the Goulburn Valley region of Victoria, Australia (April 2021) using (i) serial dilution plating on selective media and (ii) the insect-bait method [47]. 12 soil samples were collected across the sampling site (1.25 ha) at a depth of 10–15 cm, using a manual core sampler. Each sample consisted of 20 subsamples (approximately 120 g of soil) and was transferred into polyethylene bags, sealed, and stored in a portable cooler in the field. The sampler was sterilised (80% ethanol) after each collection to avoid cross-contamination. All samples were transported to the laboratory and stored at 4˚C for further processing. For easier handling, the samples were sieved through a 2 mm sieve, and stones and organic debris were removed. For the serial dilution plating, 5 g samples of soil were suspended each in 45 mL of sterile 0.05% Tween-80 and shaken for 5 min at room temperature. The suspension was diluted in series to $10^{-4}$ in sterile 0.05% Tween-80, and then 100 μL of the $10^{-3}$ and $10^{-4}$ suspension were plated in triplicate on selective media containing 1% peptone, 1% glucose, 1.25% agar, and the antibiotics chloramphenicol (50 mg/l) and cycloheximide (400 mg/l). The plates were incubated at 26˚C for 14 days. Colonies were sub-cultured on Sabouraud's dextrose agar (SDA) (Oxoid, Thermo-Fisher Scientific, United Kingdom) to obtain pure cultures. For the insect-baiting method, we used wax moth larvae (*Galleria mellonella*) and mealworm larvae (*Tenebrio molitor*). 36 soil samples (200 g) were baited with either five, fifth-instar wax moth larvae or five freshly shed mealworm larvae, using published methods [48] and incubated at 26˚C and 70% RH. After 13 to 17 days, dead larvae were removed from the soil, surface sterilised (70% ethanol for 30 s, then rinsed three times in sterile distilled water), transferred to Petri dishes lined with sterile dampened filter paper, and incubated at 26˚C. Cadavers were monitored daily and fungal colonies were isolated from fungal hyphae emerging from the cadavers cultured on SDA agar, as described previously. Cultures were initially identified using morphological keys [49, 50]. To compare isolates obtained from the Goulburn Valley region, we also tested two *Metarhizium*

**Table 1. Details of entomopathogenic fungi evaluated against preimaginal life stages of Queensland fruit fly.**

| Isolate Code | Identification | Isolate host | Origin |
|---|---|---|---|
| BGV1 | *Beauveria bassiana* | Soil, selective agar | Goulburn Valley Region (Merrigum), VIC |
| BQ1 | *B. bassiana* | *Musca domestica* | QLD |
| MQ1 | *Metarhizium anisopliae* | Soil | South Johnstone, QLD |
| MQ2 | | | |
| MQ3 | | | Aratula, QLD |
| MGV1 | *M. anisopliae* | Soil, *Galleria mellonella* | Goulburn Valley Region (Merrigum), VIC |
| MGV2 | | | |
| MGV3 | | | |
| MGV4 | *M. anisopliae* | Soil, selective agar | |
| MGV5 | | | |
| MGV6 | | | |
| MGV7 | | | |
| MGV8 | | | |
| MGV9 | | | |
| MM1 | *M. anisopliae* | Soil, *Tenebrio molitor* | Mallee Region (Mildura), VIC |
| MM2 | | | |

isolates obtained from soil in an almond block in the Mallee region of Victoria (collected as part of a separate study, August 2021), and one *Beauveria bassiana* and three *Metarhizium* isolates obtained from Queensland (Table 1).

Genomic DNA was extracted by the chemical lysis method using the DNeasy® Plant Pro Kit (QIAGEN). The internal transcribed spacer (ITS) region was amplified using ITS5 (5'– GGAAGTAAAAGTCGTAACAAGG) and ITS4 (5'–TCCTCCGCTTATTGATATGC) primers [51]. Additionally, the translation elongation factor 1-alpha (EF1-α) region was also amplified using the primers 983F (5'–GCYCCYGGHCAYCGTGAYTTYAT) and 1567R (ACHGTRCCRATAC CACCRAT) [52]. The PCR master-mix was prepared according to the Qiagen AllTaq PCR core kit, with 25 μL volume reactions and 3 μL of gDNA each. Amplifications were carried with the following PCR cycling conditions after an initial denaturation step at 95°C for 2 minutes, followed by 35 cycles of (i) denaturation at 95°C for 5 seconds, (ii) annealing at 58°C for 15 seconds, and (iii) extension at 72°C for 10 seconds, and a final extension step at 72°C for 2 minutes. The quality and size of a 5 μL PCR product were visualised on 1% agarose gel, and PCR products were sequenced by Macrogen (South Korea). Phylogenetic analysis was carried out for the isolates using MEGA 11 [53]. A preliminary phylogenetic analysis was conducted for the TEF1-α sequences of *Metarhizium* spp. using the software Geneious Prime version 2023.2.1 (Biomatters). A Tamura-Nei genetic distance model was conducted using the Neighbor-joining method with bootstrap resampling of 1000 replicates. The TEF1-α nucleotide sequences were submitted to GenBank (NCBI) and accession numbers obtained for each isolate in the phylogenetic tree (S1 Fig). All isolates were identified as species within the *Metarhizium anisopliae* complex and are hence described as *Metarhizium anisopliae sensu lato* (*M. anisopliae* s.l.).

## Initial screening for pathogenicity of isolates against third-instar larvae

Two *B. bassiana* and 14 *M. anisopliae* s.l. isolates (Table 1) were screened against third-instar larvae of Queensland fruit fly. All isolates were grown on SDA for 21 days. Conidia were harvested from sporulating cultures by adding sterile 0.05% Tween-80 and scraping spores with a sterile cell spreader. The resulting suspension was filtered through sterile Miracloth (22–

25 μm; Millipore, Sigma-Adrich Pty. Ltd.) into a Falcon tube and vortexed for 1 min to homogenise. The concentration was calculated using a Neubauer Improved hemocytometer and adjusted to achieve a target application rate of $1 \times 10^8$ conidia $g^{-1}$ vermiculite. The viability of conidia was evaluated for each isolate by plating a $10^{-6}$ dilution on SDA. Colony forming units (CFUs) were counted after seven days, and the viability was estimated [48]. The bioassay units consisted of transparent plastic vessels (946 mL, SteriCon™ 13, Austratec) containing 10 g of vermiculite (grade 1) each. The adjusted fungal suspensions were applied to each respective container by drenching with a pipette onto the vermiculite to achieve 33% total water content. Inoculated vermiculite was subsequently mixed thoroughly in each container to ensure an even distribution of inoculum. The control was treated with sterile 0.05% Tween-80 by the same procedure. To each container, a 60 mm Petri dish containing Queensland fruit fly-infested carrot diet (26 g diet with approximately 39 eggs) was placed onto the vermiculite. Larvae were able to exit the diet to pupate in the treated substrate. The containers were covered with perforated lids (nine 1 mm holes) to allow gas exchange and incubated at 27˚C, with 60% RH and a 12:12 (L:D) photoperiod [36]. After 14 days of incubation, emerged flies were counted. Unemerged pupae were recovered and surface sterilised in 80% ethanol for 30 s, followed by three rinses with sterile distilled water. Surface sterilised pupae were subsequently transferred onto 1% agar containing 0.5 g chloramphenicol / L for incubation to assess mycosis.

## Assessing and comparing growth of fungal isolates

A series of experiments were conducted with three *M. anisopliae* s.l.). isolates (MGV1, MQ1, MM2) selected based on the initial pathogenicity assessment (Table 1). The further aim was to compare these isolates for differences in hyphal growth, conidial germination, conidial yield, viability, and pathogenicity at different temperature regimes. To compare hyphal growth, conidial suspensions of $1 \times 10^7$ conidia $mL^{-1}$ were prepared as previously described from 21-day old sporulating cultures of the three *Metarhizium* isolates. 100 μL of each suspension was spread onto SDA media and incubated at 27˚C for three days. A 5 mm mycelial mat was cut using a sterilised cork borer and transferred onto the centre of a fresh SDA plate for each isolate. The plates were previously marked with X and Y axes on the underside and then incubated at 21, 27, and 33˚C, respectively. The radial growth of each fungus was measured along the axes every second day for 21 days, with ten replicates per treatment for both the isolate treatment and temperature level. The maximum radial growth was reached when the isolate filled the Petri dish. Conidial yield was assessed for each isolate after 21 days by cutting three 100 mm circular disks with a sterile cork borer and suspending each in 10 mL sterile 0.05% Tween-80. The suspension was mixed using a vortex for 1 min and the quantity of conidia was estimated as conidia $mL^{-1}$ using a haemocytometer. Each isolate and temperature level were replicated in triplicate. The germination rate of each isolate was assessed by plating 20 μL of $1 \times 10^6$ conidia $mL^{-1}$ suspension on a thin film of SDA prepared on a glass slide. The glass slide was covered with a cover slip and incubated in a Petri dish on moist filter paper at the three temperature levels. Germination was quantified after 8, 10, 14, 18, and 24 hours by counting the first 100 conidia seen; this was replicated three times across different fields of view and for three slides per isolate. The conidia were considered germinated when the germ tube was twice the length of the conidium width [48, 54]. Additionally, spore viability was determined by plating 100 μL of the conidial suspension, diluted to $10^{-6}$ on SDA, incubating at 21, 27, and 33˚C, respectively, and counting CFUs after 5 days [26, 55]. To assess the virulence of the three isolates at different temperatures, groups of five third-instar larvae were immersed in 1 mL of each suspension, prepared as previously described, at a concentration of $1 \times 10^7$ conidia $mL^{-1}$

and subsequently transferred onto 10 mL of sterile vermiculite (33% water content). A control group was immersed in sterile 0.05% Tween-80. After 2 days, the pupated larvae were transferred onto 1% agar (as described above) to assess mycosis. This bioassay was conducted twice with four replicates per isolate.

### Pathogenicity against third-instar larvae and pupae of different ages

Bioassays were conducted to investigate the most susceptible juvenile stage of Queensland fruit fly, using the three candidate isolates MQ1, MGV1, and MM2, and a no-inoculum control, applied to sterile vermiculite as a pupation substrate according to the method previously described, except that substrate was prepared with a concentration of $9 \times 10^7$ conidia $g^{-1}$ vermiculite for the isolate treatments. Across six treatments, groups of 20 third-instar larvae, or 20 pupae with treatments at different ages (1, 3, 5, 7, and 9-day old) were transferred onto the treated vermiculite. Larvae were allowed to burrow naturally into the substrate, while pupae were carefully mixed into the vermiculite with a sterile hoop to ensure exposure to the respective substrate treatment. All bioassay containers were incubated as described previously and arranged in a complete randomised nested split-plot design with three replicates per treatment, the whole bioassay was repeated three times (n = 9). Units were observed daily and the number of flies that emerged was recorded. Pupae that failed to emerge were carefully recovered, surface sterilised, and incubated as previously described to assess mycosis. Mortality by mycosis was verified after 3 to 5 days by observation of hyphal growth, and then after 7 to 10 days by observation of sporulating cadavers. In addition to estimating the viability of the inoculum by quantifying the number of CFUs of the original suspension, the distribution of the inoculum in the vermiculite was assessed by suspending 1 g of inoculated vermiculite in 10 mL of 0.05% Tween-80 from each experimental container. A $10^{-6}$ dilution was then prepared and plated onto SDA with three replicate plates per unit. After five days, the number of CFUs was counted to enable the calculation of viable spores per $g^{-1}$ volume of vermiculite.

### Survival of emerging adults

To investigate the susceptibility of emerging Queensland fruit fly adults to inoculated substrate, 12-day old pupae that were expected to emerge within 24 hours were treated as previously described in the pupal treatments. Bioassay containers were monitored after 8, 12, 16, and 24 hours, and any emerged flies were directly transferred into separate containers and supplied with water and sugar for nutrition. Flies were held at room temperature, and life span was recorded daily until 11 days after emergence. Cadavers were surface sterilised and incubated as previously described to access mycosis (Fig 1).

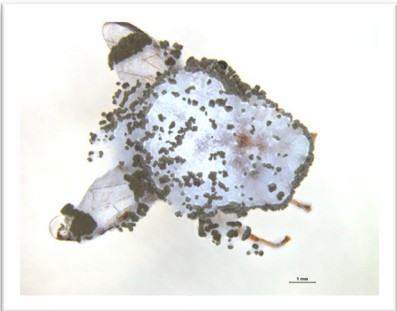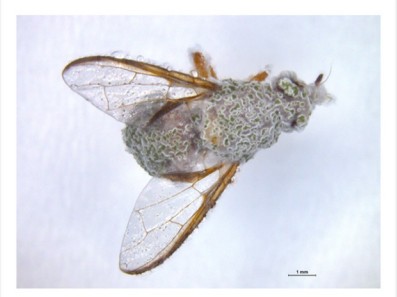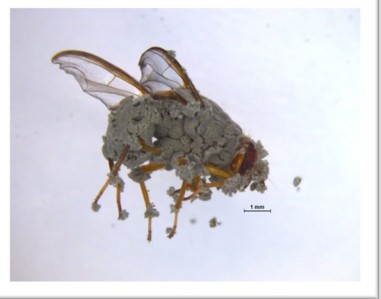

**Fig 1. Photographs depicting mycosis of adult Queensland fruit fly, *Bactrocera tryoni*, inoculated each with three *Metarhizium anisopliae sensu lato* isolates, from left to right: isolates' MM2, MGV1, and MQ1.**

### Dose response in non-sterile orchard soil

This final bioassay investigated the concentration of inoculum required for fungal entomopathogens to infect third-instar larvae and achieve satisfactory rates of mortality, when exposed to non-sterile orchard soil. Soil was collected from the Agriculture Victoria Smart-Farm at Tatura (Victoria), sieved through a 2 mm soil sieve to remove stones, organic debris, and insects, and then mixed with vermiculite in a 1:1 volume ratio for use as a pupation substrate in bioassays. The presence of resident fungal entomopathogens was first assessed with 15 soil samples according to the method described previously for isolating fungi from soil. *Metarhizium*, *Beauveria*, *Isaria*, *and Lecancillium* species were detected in the assessed soil samples, although the quantity was generally low. The isolate MQ1 was prepared as an aqueous suspension of conidia in 0.01% Tween-80 and adjusted to achieve the following concentrations: 1 x $10^6$, 2.5 x $10^6$, 5 x $10^6$; 1 x $10^7$, 2.5 x $10^7$, and 5 x $10^7$ conidia $g^{-1}$ of soil, with 28% soil moisture content. A no-inoculum control was prepared with 0.01% Tween-80. Gravimetric soil moisture content was calculated by subtracting the weight of dry soil from the weight of wet soil, and then dividing by the weight of dry soil on the first day of the experiment and upon the emergence of adult flies. The distribution of the inoculum in soil was also assessed as previously described by plating dilutions on selective agar media. The bioassay was incubated and arranged as described above with five replicates per treatment, and the entire bioassay was repeated three times (n = 15).

### Statistical analysis

A probit model was used to analyse proportion mortality data of the initial pathogenicity screening and dose response assay. Specifically, a binominal generalised linear mixed model with a probit-link function was conducted using package lme4 [56] in R (R x64 4.2.2; RStudio). The fungal isolate factor was treated as a fixed effect in the model, and the bioassay experiment number was treated as a random effect, to account for any variation arising between experiments. Model assumptions were inspected by plotting residuals versus fitted values, and by testing for overdispersion based on Pearson's chi-squared residual test. Furthermore, a *post hoc* Tukey honesty significance differences (HSD) test was performed using a generalised linear hypotheses multiple comparison procedure using the package multcomp [57]. Mean percent mortalities were graphed with the package ggplot2 [58] to visualise and present the treatment effects. An ANOVA model or non-parametric Kruskal-Wallis (KW) test within the package agricolae [59] was used to analyse the percent mortality data of the life stage assay and the data of the temperature assay after testing the data for normal distribution and homogeneity of variances. Because older pupae ages (3-, 5-, 7-, and 9-day old) did not show mortality through fungal infection, the data was removed for statistical analysis. Percent mortality data, temperature data, and trends of fungal germination were plotted over time with the package ggplot2. Viable rates of conidia were analysed with a regression model. A cox proportional-hazards model conducted with the packages survival and survminer [60] was used to analyse the survival of emerging adults after exposure to the fungal isolate treatment. A multiple log-rank test was used to compare survival curves by treatment group and plotted using the package survival. For raw data see S1 Appendix.

## Results

### Initial pathogenicity screening of isolates against third-instar larvae

The two *B. bassiana* and 14 *M. anisopliae* s.l. isolates tested for pathogenicity against third-instar larvae showed significant differences in proportional mortality compared to the no-

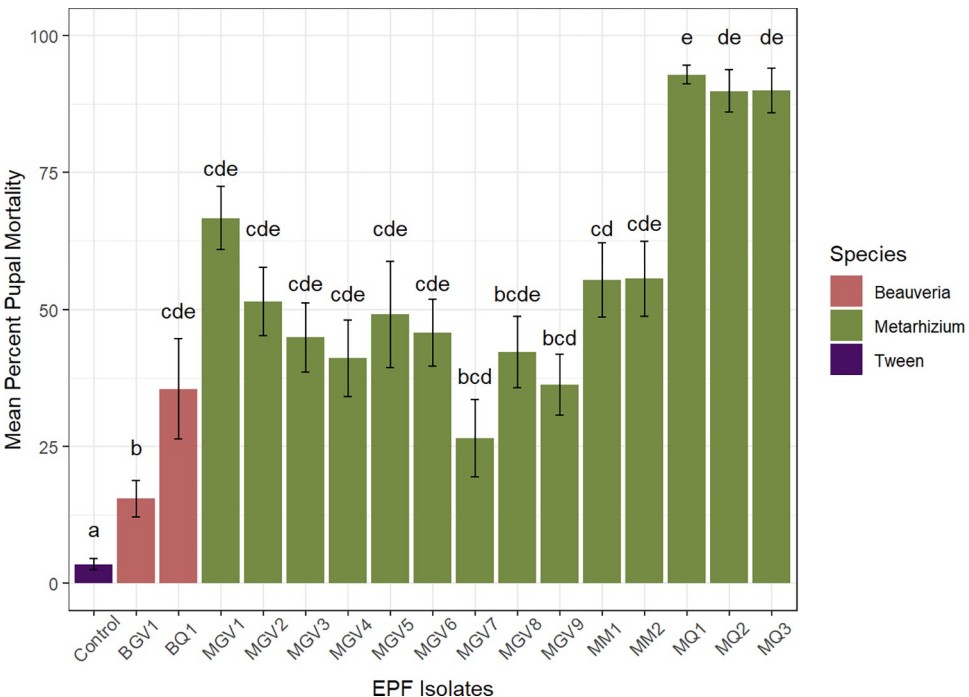

**Fig 2. Initial pathogenicity of 16 isolates tested (at 9 x 10$^7$ conidia g$^{-1}$) against third-instar larvae of Queensland fruit fly.** Bars represent the mean percentage mortality (± standard error). Different letters denote significant differences (p < 0.05) analysed on proportion data with generalised linear mixed model analysis with post-hoc Tukey HSD test.

inoculum control ($F_{(16)} = 24.2; P < 0.001$), based on the probit model analysis (S1 Table). *M. anisopliae* s.l. isolates MQ1, MQ2, and MQ3 were the most pathogenic to Queensland fruit fly followed by MGV1, MM2, MM1, and MGV2 (Fig 2). Overall, the *Beauveria* isolates were less pathogenic compared to the *Metarhizium* isolates; although only *M. anisopliae* s.l. isolates MQ1, MQ2, MQ3, and MGV1 were significantly different from the *B. bassiana* isolates and no inoculum-control. The viable conidial concentration applied differed between the isolates tested ($\chi^2_{(16)} = 239.3; P < 0.00; KW$) and the bioassays conducted ($\chi^2_{(12)} = 72.7; P < 0.01; KW$). However, the viable conidial concentrations applied were still within high rates of 9.29 x 10$^6$–1.55 x 10$^8$ conidia g$^{-1}$ of vermiculite between the isolates tested.

## Growth of selected fungal isolates

Radial growth rates were significantly different among temperature treatments (21˚C, 27˚C, and 33˚C) for the isolates tested ($\chi^2_{(8)} = 85.5; P < 0.01; KW$). Isolate MQ1 reached maximum radial growth by 21 days at 21˚C and 27˚C and performed better at the higher temperature of 33˚C compared to MGV1 and MM2 (Table 2), whereas MGV1 only reached maximum growth at 27˚C. Overall, MM2 showed the lowest radial growth at all temperature levels tested. The conidia yield per 10 mm agar disc after 21 days showed significant differences between the isolates ($\chi^2_{(8)} = 24.9; P = 0.002; KW$). The highest yield was recorded for isolate MQ1 at all three temperature levels, followed by MM2 and MGV1 (Table 2). Germination of conidia was significantly different among isolates when measured after 14 hours ($\chi^2_{(8)} = 24.6; P = 0.002; KW$), with conidia of MQ1 germinating more rapidly with increasing temperatures (Fig 3), whereas MM2 did not show a difference in germination rate between 27˚C and 33˚C (Table 2). MGV1 showed a significantly lower proportion of germinated conidia at 33˚C compared to 27˚C.

**Table 2. Mean values for growth attributes and virulence of the three *M. anisopliae* s.l. isolates MQ1, MM2, and MGV1 at different temperatures.** Data represents means of replicates ± standard error. Treatments followed by different letters indicate significant differences ($P < 0.05$) based on non-parametric Kruskal-Wallis (KW) of percent data followed a Bonferroni post hoc analysis.

| | 21°C | | | | 27°C | | | | 33°C | | | |
|---|---|---|---|---|---|---|---|---|---|---|---|---|
| | MQ1 | MM2 | MGV1 | 0.05% Tween-80 | MQ1 | MM2 | MGV1 | 0.05% Tween-80 | MQ1 | MM2 | MGV1 | 0.05% Tween-80 |
| **Final colony diameter after 21 days [mm ± SE]** | 89.20 ± 0.67 a | 49.67 ± 0.32 d | 71.19 ± 1.48 c | n/a | 90.00 ± 0.00 a | 69.11 ± 1.09 c | 90.00 ± 0.00 a | n/a | 75.78 ± 0.57 b | 32.51 ± 0.52 e | 35.57 ± 1.79 e | n/a |
| **Conidia yield after 21 days [x 10$^7$ conidia mL$^{-1}$ ± SE]** | 5.64 ± 1.33 a | 1.27 ± 0.18 cd | 0.41 ± 1.23 de | n/a | 8.36 ± 0.81 a | 4.19 ± 0.60 ab | 2.16 ± 1.14 bc | n/a | 2.38 ± 0.46 bc | 0.00 ± 0.00 e | 0.25 ± 0.66 de | n/a |
| **Proportion germinated conidia after 14 h ± SE** | 0.40 ± 0.07 bc | 0.11 ± 0.04 d | 0.08 ± 0.01 d | n/a | 0.67 ± 0.02 a | 0.33 ± 0.01 bc | 0.61 ± 0.03 ab | n/a | 0.99 ± 0.01 a | 0.33 ± 0.04 c | 0.27 ± 0.01 cd | n/a |
| **Viability [%] after five days** | 99 | 80 | 82 | 0 | 88 | 98 | 100 | 0 | 99 | 0 | 4 | 0 |
| **Pupal mortality [% ± SE]** | 75 ± 3.27 ab | 90 ± 3.78 a | 43 ± 9.60 bc | 13 ± 13 de | 90 ± 6.55 a | 53 ± 8.40 bc | 23 ± 5.90 cd | 0 ± 0.00 e | 100 ± 0.00 a | 0 ± 0.00 e | 25 ± 3.27 cd | 0 ± 0.00 e |

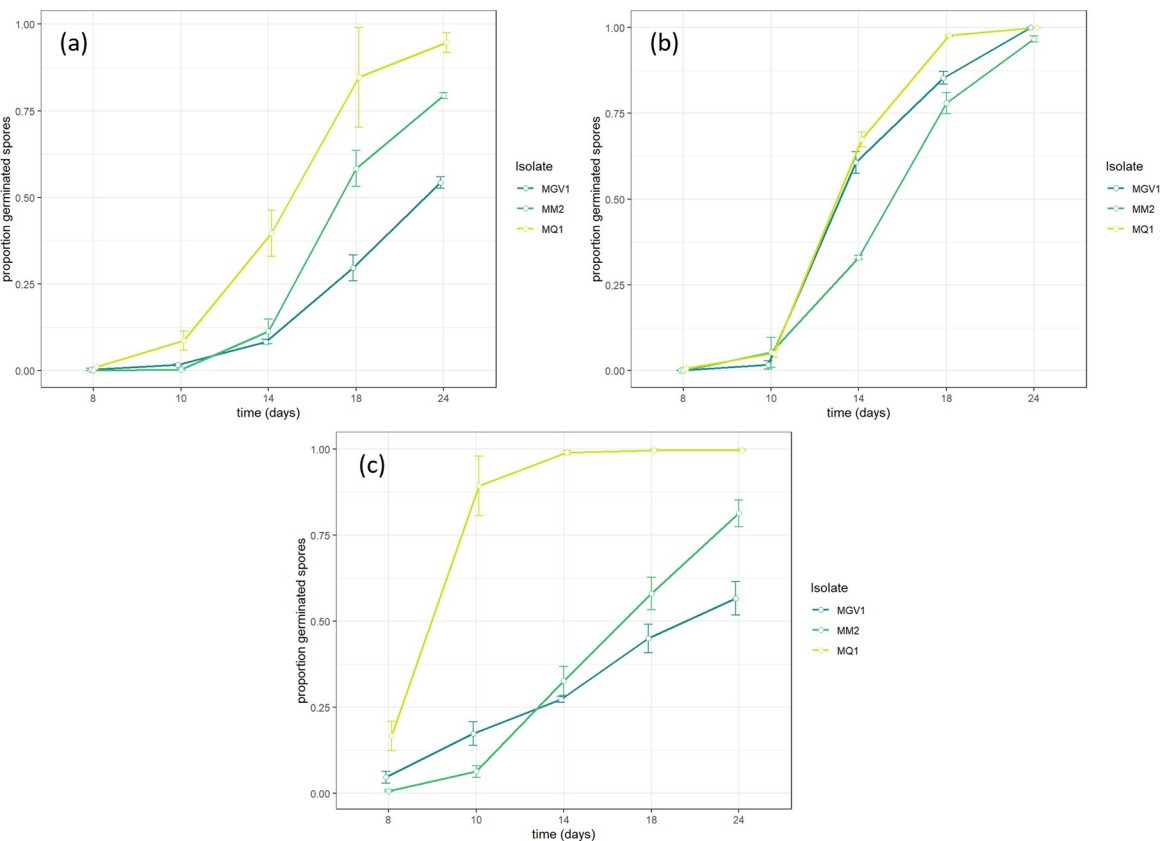

**Fig 3.** Proportion of germinated conidia over 24 hours for MQ1, MM2, and MGV1 at different temperatures: (a) 21˚C, (b) 27˚C, and (c) 33˚C.

## Effect of temperature on pathogenicity of selected isolates

Differences in virulence towards third-instar larvae were recorded among isolates and temperature levels ($\chi^2_{(11)}$ = 80.7; *P < 0.001; KW*). At 21˚C, *M. anisopliae* s.l. isolate MM2 showed a higher percentage of dead mycosed pupae, followed by MQ1. At 27˚C, MQ1 performed significantly better, followed by MM2. The *M. anisopliae* s.l. isolate MGV1 showed the lowest percentage of mycosed pupae at 21˚C, with no significant difference observed at 27˚C and 33˚C, compared to MM2 and the untreated control. By contrast, MQ1 resulted in significantly higher insect mortality at 33˚C, while MM2 did not cause any fungal infection at 33˚C (Table 2). The percent viability calculated by CFUs ranged between 80–100% depending on the isolate and temperature level tested. It is worthwhile noting that MM2 and MGV1 showed low viability of 0% and 4%, respectively, at 33˚C. Only MQ1 had viable conidia of 99% at 33˚C after five days (Table 2). The plates of MM2 and MGV1 were incubated further at 27˚C for 5 days, and at this temperature, both isolates achieved viable rates of 82% (MGV1) and 99% (MM2).

## Pathogenicity of selected isolates against different juvenile life stages

The pathogenicity of *M. anisopliae* s.l. isolates MQ1, MM2, and MGV1 was investigated against third-instar larvae and pupae of different ages. Third-instar larvae were found to be the most susceptible group ($\chi^2_{(3)}$ = 31.6; *P < 0.001; KW*) followed by one-day old pupae ($\chi^2_{(3)}$ = 9.7; *P = 0.021; KW*). Isolate MQ1 achieved 98% mortality when third-instar larvae were

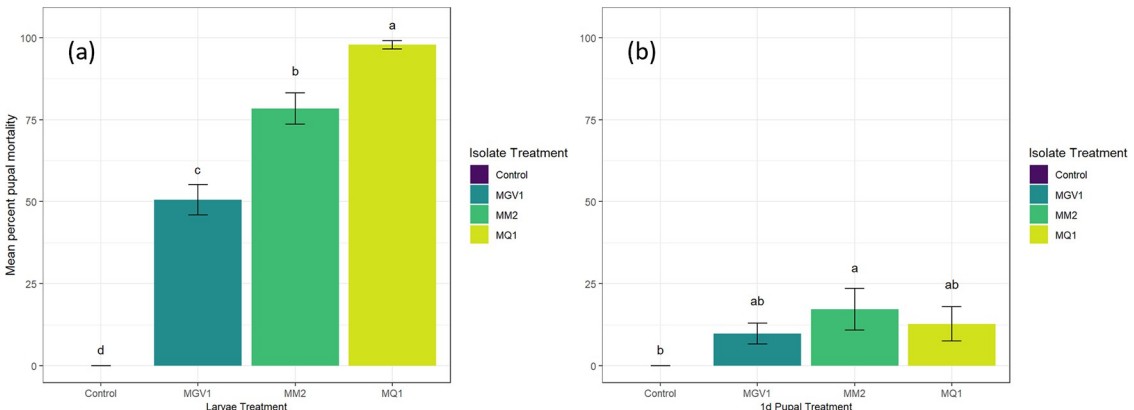

**Fig 4.** Pathogenicity of isolate MQ1, MM2 and MGV1 in vermiculite ($9 \times 10^7$ conidia $g^{-1}$) against (a) third-instar larvae and (b) one-day old pupae of Queensland fruit fly. Controls were treated with sterile 0.05% Tween-80. Bars represent the mean percent mortality (± standard error). Different letters denote significant differences between isolate treatments and the no-inoculum control.

exposed to inoculated vermiculite; followed by MM2 and MGV1 with 78% and 51% mortality, respectively (Fig 4). Percent conidial viability for the larvae treatments were 97%, 94%, and 95% for MQ1, MM2, and MGV1. One-day old pupae were significantly less susceptible compared to the larval stage, with only 17%, 13%, and 10% pupal mortality for isolates MM2, MQ1, and MGV1, respectively (Fig 4). Again, isolates showed high conidial viability of 97% for MQ1, 94% for MM2, and 95% for MGV1.

## Survival of treated emerging adults

Emerging Queensland fruit fly adults were susceptible to all inoculum treatments, with significant differences found between isolates (Fig 5). The Cox proportional-hazards model used to

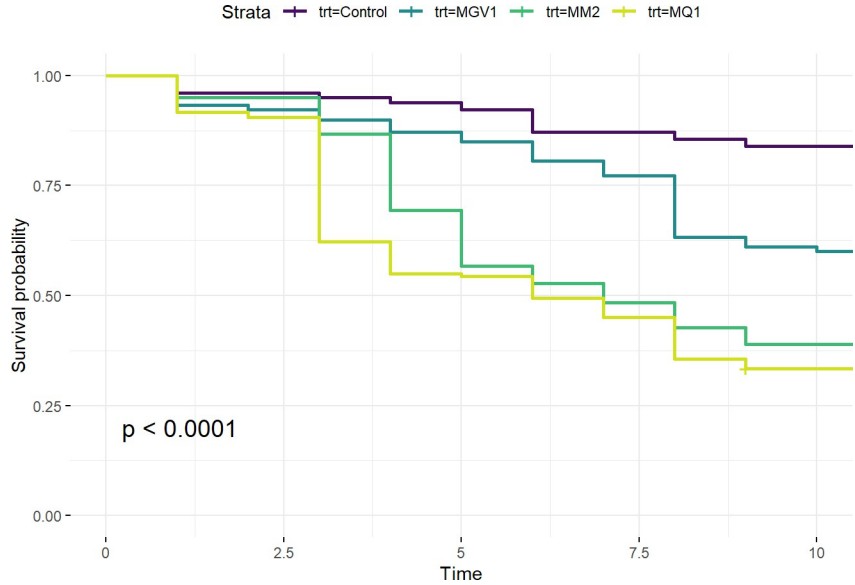

**Fig 5. Survival curves of adult Queensland fruit fly emerged from fungal isolate treated vermiculite ($9 \times 10^7$ conidia $g^{-1}$) showing the time-mortality response.** Survival curves are analysed using Cox proportional-hazards model followed by different letters denote significant differences ($p < 0.05$) with a multiple comparison log-rank test.

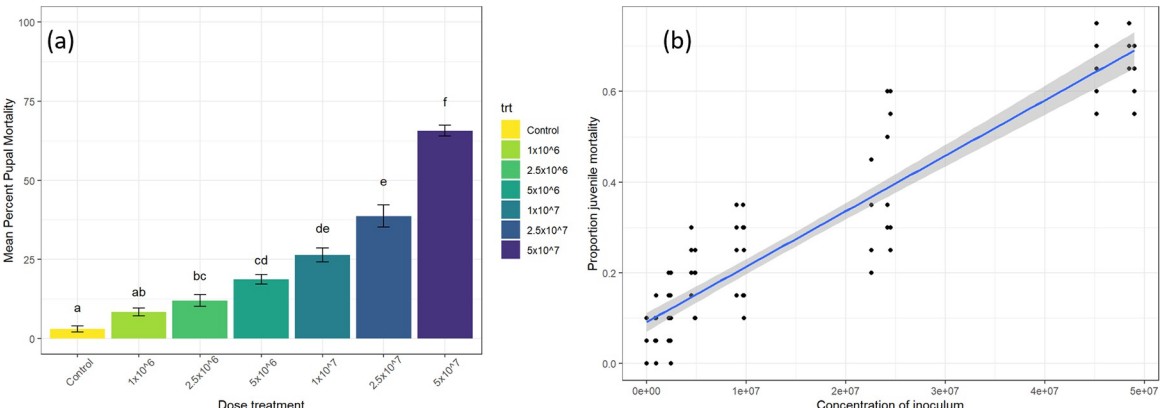

**Fig 6.** (a) Mean percentage pupal mortality of the isolate MQ1 towards third-instar larvae of Queensland fruit fly in non-sterile orchard soil. Data represents means of replicates ± standard error. Treatments followed by different letters indicate significant difference ($p < 0.05$) through generalised linear mixed model analysis of proportional data followed a Tukey HSD test. (b) Linear regression for proportion juvenile mortality with 95% confidence interval indicated in grey ($9.127^{-02}$+($1.223^{-08}$×*viable rate applied*)+0.084).

investigate the survival probability of adult flies over 10 days indicated that *M. anisopliae* s.l. isolate MQ1 and MM2 were significantly more virulent in the first 10 days after emergence, compared to MGV1 and the untreated control (*likelihood ratio test = 132.8; df = 3; p < 0.00*). The percent mortalities after 9 days were 16.9%, 56.3%, 66.1%, and 75.7% for the untreated control, MGV1, MM2, and MQ1, respectively. The conidial viability of the inoculum was 97% for MQ1, 89% for MM2, and 83% for MGV1.

## Dose response in non-sterile orchard soil

When MQ1 was applied to non-sterile soil, the linear regression model indicated a significant positive relationship between the proportion of mycosed pupae and increasing fungal concentration applied to mortality (adjusted $R^2 = 0.85; F(1, 103) = 581.1; P < 0.01$). The slope of the fit between the viable rate applied to the proportion mortality rate was $1.223^{-08}$ with an intercept of $9.127^{-02}$, indicating that a tenfold increase in concentration had ~11% increase in mortality. Concentrations higher than $2.5 \times 10^6$ conidia g$^{-1}$ soil showed significant differences in pupal mortality compared to untreated control ($F_{(6)} = 37.4; p < 0.001$). The highest percent pupal mortality of 66% mortality was achieved with the highest concentration of $5 \times 10^7$ conidia g$^{-1}$ soil (Fig 6). Between the experiments conducted, the viability of the isolate MQ1 ranged between 90 98%. The soil moisture content was consistent throughout the experiment with minimal moisture loss of 4.1%. Moreover, the fungal concentration gram$^{-1}$ of dry soil observed was within the range of the estimated dosage applied.

## Discussion

Isolates of *B. bassiana* and *M. anisopliae* s.l. species were evaluated for pathogenicity in soil-dwelling juvenile stages of Queensland fruit fly, to explore the potential use of these fungi as biopesticide agents. In initial screening trials, juvenile life stages were found to be highly susceptible to infection by the two fungal genera tested. *M. anisopliae* s.l. isolates showed higher pathogenicity to vulnerable life stages of Queensland fruit fly compared to the *Beauveria* isolates, with significant differences observed among isolates. Mortality rates in our study were comparable to those in studies screening entomopathogenic fungi against other tephritid fruit fly species [29–31, 34, 35, 40, 61]. For example, pathogenicity studies using *M. anisopliae* against larvae of carambola fruit fly (*B. carambolae*) demonstrated mortality of 70% in pre-

pupating larvae in sterile soil [38]. A study screening 20 different isolates of *M. anisopliae* against third-instar larvae of Mexican fruit fly (*Anastrepha ludens*) demonstrated mortalities ranging from 37.5–98.75% [26].

Our results showed mortality varying from 26–93% depending on the tested *Metarhizium* isolate. Both of the other studies mentioned above used the same high fungal concentration, while the application method differed between the studies. Our study incorporated the fungal suspension in the substrate, whereas these other two studies sprayed the suspension into the soil or applied the fungal entomopathogen through immersion of the larvae in a fungal suspension. Our method might therefore better represent the pest ecology and intended application environment [16].

Pathogenicity can vary between fungal species, strains, and even isolates [62]. For example, screening conducted against peach fruit fly (*B. zonata*) and Oriental fruit fly (*B. dorsalis*) demonstrated that *B. bassiana* was more effective compared to *Metarhizium* isolates tested [35, 40], which differs from our findings on Queensland fruit fly. Fungal species and strain selection are therefore important to consider in terms of the future development of fungal entomopathogens to control a particular fruit fly pest [16].

Our study is the first to test the susceptibility of soil-dwelling life stages of Queensland fruit fly (third-instar larvae and pupae) following exposure to *M. anisopliae* s.l. isolates. Our initial pathogenicity study revealed the susceptibility of the soil-dwelling life stages of Queensland fruit fly to the isolates tested. However, the study provided no indication of when infection in soil occurred, the duration of the incubation period, and which life stage (larvae, pupae) was susceptible to infection. Thus, we also tested the susceptibility of different age groups in order to better understand when infection can occur. Pre-pupating third-instar larvae were the most susceptible soil-dwelling life stage to entomopathogenic fungi, which is consistent with what has been observed in other tephritid fruit fly species [30, 35, 40, 62, 63]. Our findings showed that one-day old pupae were comparatively less susceptible than larvae but were nonetheless vulnerable. These findings differ from a study on peach fruit fly (*B. zonata*), which did not observe significant mortality when fresh pupae were inoculated by spray or inoculated in sterile soil with *B. bassiana*, *M. anisopliae*, or *Isaria fumosorosea* in different concentrations (1 x, 2 x, and 3 x $10^8$) [40]; however, the age of the pupae was not precisely determined in that instance.

Older pupal ages did not show any sign of fungal infection in our study. The greater melanisation and sclerotisation of the mature pupal cuticle may prevent or limit the ability of fungal entomopathogens to infect older pupae [64]. This indicates a limited period for the pre-pupal stage to encounter sufficient fungal spores to succumb to infection. By contrast, significant mortality has been demonstrated when 4 to 5-day old pupae of the peach fruit fly (*B. zonata*) and Oriental fruit fly (*B. dorsalis*) are exposed to *B. bassiana* isolates or *M. anisopliae* isolates in sterile soil treated with concentrations of 1 x $10^7$ and 1 x $10^8$ conidia mL$^{-1}$ [35]. *Beauveria* isolates caused higher mortalities in both fly species compared to the *Metarhizium* isolates. However, the study used the cumulative mortality of pupae and adults' post-emergence when pupae were treated. The results may therefore have been confounded by the emerging adults becoming infected during pupal eclosion. We also observed the mortality of adult flies emerging from treated pupae, and subsequently investigated the infection of new adults by exposing 12-day old Queensland fruit fly pupae. Adults emerging from this pupal treatment had a significant decrease in survival within the first 10 days. Mortalities ranged between 56–76%, depending on the isolate tested. By comparison, within five days post-emergence, 100% adult mortality was reported when the late instar larvae of the carambola fruit fly (*B. carambolae*) were exposed to *M. anisopliae* [38]. The longer time of exposure to the inoculum from the late instar larval stage until post-emergence is likely to cause higher mortalities observed compared

to what we observed in this present study. The soil-dwelling life stages of Queensland fruit fly can therefore be targeted when entering the soil as larvae and exiting the soil during adult emergence.

As tephritid fruit flies emerge as sexually immature adults, this provides an opportunity to kill adults before they reach maturation. *Bactrocera* species reach sexual maturity around 10 days after emergence under laboratory conditions [4]. For Queensland fruit fly, maturation periods from 5 to 31 days have been reported, depending on rearing history and conditions [4], although maturation periods under natural conditions may vary. We provided evidence that the tested *Metarhizium* isolates resulted in a significant reduction in adult survival within the first 10 days following exposure to the inoculum, if no infection occurs prior to pupation. Adults that survive treatment upon maturation may show a reduction in fecundity, as observed in females of Mexican fruit fly (*A. ludens*) were treated with *B. bassiana*, and females of Mediterranean fruit fly (*C. capitata*) when exposed to *M. anisopliae* [28, 33].

Environmental temperature can strongly influence the efficacy, longevity, growth, spore production, germination, pathogenicity, and virulence of fungal entomopathogens towards different hosts and has proven to be a critical factor for their success in the field [43, 44, 65–67]. Our findings demonstrated that the *M. anisopliae* s.l. isolate MQ1 had the highest potential at the temperature levels tested by achieving the highest radial growth rate, the highest germination rate after 14 hours, with the highest conidial yield, in addition to high conidial viability, when measured by the number of CFUs. Generally, the optimum temperature for fungal entomopathogen growth and germination ranges between 25–30˚C [68]. Studies often investigate the viability of a specific isolate by simply counting germinated spores [68–71]. Besides the germination rate, however, we also investigated the viability of isolates by quantifying the number of CFUs that were able to grow at the temperature tested. At 33˚C, only MQ1 showed viable colonies, whereas MM2 and MGV1 did not, even when the isolates showed germination rates over 50% after 24 h. Interestingly, when the cultures were moved to 27˚C, these isolates resumed growth and showed high viability by CFU counts. These observations were comparable with a study that tested the upper temperature limits of *Metarhizium* species [72]. After exposure to high temperatures for 10 days, some isolates were able to resume growth when transferred to lower temperatures. This indicates that testing the germination rate might not be sufficient to report on an isolate's viability.

It is widely recognised that high temperatures have a negative effect on conidial germination and viability [71, 73, 74] but the decrease in fungal efficiency at high temperatures is also possibly associated with increased stress responses of fungal isolates [75, 76] or due to the thermal response of the insect host [77, 78], rather than conidia germination and fungal growth rate directly. The current study assessed the virulence towards third-instar larvae of Queensland fruit fly. Again, *M. anisopliae* s.l. isolate MQ1 showed the highest potential across the three temperatures tested. However, the *M. anisopliae* s.l. isolate MM2 showed slightly higher mortalities in third-instar larvae at 21˚C, compared to MQ1. Though this difference was not significant, it may indicate that MM2 might perform better at cooler temperatures compared to the other isolates, though we have not tested below the optimal temperature range in this study. Different fungal entomopathogen species and isolates can show differences in their optimal temperature range [70, 79] and thus might be more suitable for application in different geographical regions and during the seasons to maximise their efficiency in the field [43, 44]. Further work might explore regional and seasonal soil temperatures relevant in Australia, which covers a large range of climate zones, from tropical to temperate and arid regions.

Besides temperature, soil microbiota, moisture content, structure, and chemistry also influence the availability and activity of fungal spore activity, which causes variability in host-specific fungal infection in soil compared to vermiculite or other sterile substrate [44, 80]. This

host specificity can additionally be dose-dependent [81] and ultimately alter the number of spores needed; hence, the dosage needed to achieve a notable effect on a pest population. *M. anisopliae* s.l. isolateMQ1 was tested at different dosages in non-sterile orchard soil. Our results revealed that MQ1 achieved up to 66% pupal mortality when third-instar larvae were exposed to high concentrations of $1 \times 10^7$ conidia $g^{-1}$. Similar mortality rates of 40–83.3% were reported when *M. anisopliae* was incorporated into non-sterile soil to control Mediterranean fruit fly (*C. capitata*) [80]. By contrast, *M. anisopliae* and *M. robertsii* sprayed on non-sterile soil ($1 \times 10^8$ conidia $mL^{-1}$) were only able to achieve 28% and 44% mortality, respectively, against larvae of carambola fruit fly (*B. carambolae*) [38]. In these studies, soil water content, which is known to influence an isolate's pathogenicity [30], was maintained by periodically moistening the soil. We controlled the water content in the soil during the experiment, which was relatively constant and optimum at 26% and therefore likely had little impact on our mortality results. It is well known that fungal entomopathogens generally achieve lower mortalities in non-sterile soil compared to sterile soil; primarily, this may be attributed due to the influence of the resident soil microbiota [44]. The non-sterile orchard soil we used contained low levels of (different) fungal entomopathogens, and interactions within the soil microbiota may have occurred, which could cause some reduction in the efficiency [82–85]. However, the mycosis observed on cadavers appeared to be caused by the inoculum treatment, based on morphological characteristics.

Seasonal field trials have been conducted to evaluate the efficacy of soil inoculation with *M. anisopliae* alone and in combination with a Spinosad-based bait spray against Oriental fruit fly (*B. dorsalis*) in Kenya [11]. This study reported that the combined use of the fungal isolate and the bait spray resulted in a significant reduction in fruit infestation compared to an untreated control [11]. A recent field study further demonstrated that a combination of *B. bassiana* or *M. anisopliae* with entomopathogenic nematodes (*S. carpocapsae* or *H. bacteriophora*) applied to target third-instar larvae of the peach fruit fly (*B. zonata*) and Oriental fruit fly (*B. dorsalis*), resulted in higher mortality than when they were applied on their own [86]. These field studies both indicate that using a combination of pest management strategies, including the use of complementary biopesticides, may be necessary to effectively suppress fruit flies in an IPM approach [11, 86–88]. However, these studies additionally demonstrate that an application of entomopathogenic fungi to soil can be effective against tephritid fruit flies in field conditions, which provides support to continue to investigate entomopathogenic fungi for use against Queensland fruit fly.

We have provided evidence that *M. anisopliae* s.l. isolates show considerable potential as a soil-applied biocontrol agent against Queensland fruit fly. The use of fungal entomopathogens could be incorporated into Integrated Pest Management (IPM) and Area Wide Management (AWM) strategies to enable more holistic control of this pest, decreasing populations of Queensland fruit fly that evade current control. Soil applied formulations could have particular use for targeting larvae emerging from fruit left on the ground at the end of the season; these populations being the source of spring populations. Future work to progress the development of fungal entomopathogens as biopesticides could look at formulation, in-field persistence, in-field efficacy, and pathogenicity to non-target insects.

## Supporting information

**S1 Fig. Phylogenetic tree constructed using the Neighbor-joining methos based of partial sequences of the EF1-α region for *Metarhizium* species that were isolated during this study and compared with sequences of *Metarhizium* spp. available from Genbank (NCBI).** Genbank accession numbers are provided as labels. The outgroup species used is *Pochonia*

*chlamydosporia*.
(TIF)

**S1 Table. Estimated regression parameters, standard error, z-values, and P-value for the binomial (probit) GLMM conducted to assess proportion mortality by isolate treatment.** The estimated variance for the random effect 'bioassay' is 0.016.
(TIF)

**S1 Appendix. De-identified data set of the experimental work.**
(XLSX)

## Acknowledgments

The authors would like to thank William Boston for his support during the isolation process of fungal entomopathogens.

## Author Contributions

**Conceptualization:** Madita Prince, Aimee C. McKinnon, Diana Leemon, Tim Sawbridge, John Paul Cunningham.

**Data curation:** Madita Prince, Aimee C. McKinnon.

**Formal analysis:** Madita Prince.

**Funding acquisition:** John Paul Cunningham.

**Investigation:** Madita Prince, Aimee C. McKinnon.

**Methodology:** Madita Prince, Aimee C. McKinnon, Diana Leemon.

**Project administration:** John Paul Cunningham.

**Supervision:** Aimee C. McKinnon, Tim Sawbridge, John Paul Cunningham.

**Validation:** Madita Prince.

**Visualization:** Madita Prince.

**Writing – original draft:** Madita Prince.

**Writing – review & editing:** Madita Prince, Aimee C. McKinnon, Diana Leemon, Tim Sawbridge, John Paul Cunningham.

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
