## [Decision Letter · Decision Letter 0]

8 Dec 2023

PONE-D-23-37553Metarhizium

spp.

isolates effective against Queensland fruit fly juvenile life stages in soil.PLOS ONE

Dear Dr. Prince,

Thank you for submitting your manuscript to PLOS ONE. After careful consideration, we feel that it has merit but does not fully meet PLOS ONE’s publication criteria as it currently stands. Therefore, we invite you to submit a revised version of the manuscript that addresses the points raised during the review process.

Please submit your revised manuscript by Jan 22 2024 11:59PM. If you will need more time than this to complete your revisions, please reply to this message or contact the journal office at plosone@plos.org. Please include the following items when submitting your revised manuscript:A rebuttal letter that responds to each point raised by the academic editor and reviewer(s). You should upload this letter as a separate file labeled 'Response to Reviewers'.A marked-up copy of your manuscript that highlights changes made to the original version. You should upload this as a separate file labeled 'Revised Manuscript with Track Changes'.An unmarked version of your revised paper without tracked changes. You should upload this as a separate file labeled 'Manuscript'.

We look forward to receiving your revised manuscript.

Kind regards,

Muthugounder Subramanian Shivakumar, Ph.D.

Academic Editor

PLOS ONE

Journal Requirements:

   "MP: This research was conducted as part of a PhD fellowship with financial support from Agriculture Victoria (https://agriculture.vic.gov.au/) and La Trobe University (https://www.latrobe.edu.au). 

Work was carried out under the project: A national biocontrol program to manage pest fruit flies in Australia (Project ID: 4-EKSH327)"

Reviewers' comments:

Reviewer's Responses to Questions

**Comments to the Author**

1. Is the manuscript technically sound, and do the data support the conclusions?

Reviewer #1: Yes

Reviewer #2: Yes

2. Has the statistical analysis been performed appropriately and rigorously? 

Reviewer #1: Yes

Reviewer #2: Yes

3. Have the authors made all data underlying the findings in their manuscript fully available?

Reviewer #1: Yes

Reviewer #2: Yes

4. Is the manuscript presented in an intelligible fashion and written in standard English?

Reviewer #1: No

Reviewer #2: Yes

5. Review Comments to the Author

Reviewer #1: The authors in the manuscript entitled “Metarhizium spp. isolates effective against Queensland fruit fly juvenile life stages in soil” investigate how different fungal isolates effect against the Queensland fruit fly. This study, which aims to combat an extremely harmful species, will shed light on future studies with other species.

The aim of the article is appropriate to the scope of the journal and up-to-date. The purpose and content of the text are well-designed.

Besides the corrections in PDF, the suggestions are reported below.

• The language of the article is inadequate. The article should definitely be read by a native speaker.

• In References section, there are some mistakes. All references must be corrected according to the rules established by the journal. Species names must be written in italic. Besides, journal names should be abbreviated.

Reviewer #2: The findings of the paper definitely contribute towards control of fruit fly. However, since these are laboratory studies done in controlled environments it will be highly valuable if studies done for actual field conditions (references or work done by other researchers for similar pests) can be supplemented. In IPM approach the actual contribution of the fungi for fruit fly control cannot be augmented.

6. PLOS authors have the option to publish the peer review history of their article (what does this mean?). If published, this will include your full peer review and any attached files.

Reviewer #1: No

Reviewer #2: **Yes: **K. Yashkamal

---

## [Author Response · Author response to Decision Letter 0]

25 Dec 2023

Response to reviewer 1:

Edits have been made to address any grammatical errors where we agreed it was necessary; these changes are detailed below. Four authors are native English speakers, and we do not feel that the definite article “the” is necessary before a common name (in this case, Queensland fruit fly). We do, however, very much appreciate the thoroughness of the review provided and hope that the revised version of the manuscript is considered more than adequate with respect to the grammar and readability. The reference section was also corrected as per the requirements of the journal. 

The following comments were addressed, and changes made if necessary:

1. Removing of the space before percentage (Lines: 38, 39, 43, 44, 109,114, 115, 126, 130, 131, 132, 138, 139, 161, 170, 179, 181, 185, 186, 188, 205, 218, 219, 239, 265, 267, 268, 346, 354, 355, 356, 358, 364, 365, 366, 367, 369, 373, 382, 383, 395, 397, 399, 400, 419, 421, 422, 462, 463, 490, 515, 517, 519, and 523).

2. We have not added an article before Queensland fruit fly. Queensland fruit fly is a common name of a species and therefore does not need an article. Comments were made by the reviewer in this regard in lines’ 29, 35, 48, 54, 58, 67, 90, 93, 101, 108, 149, 168, 222, 371, 403, 411, 414, 430, 434, 465, 470, 499, 517, 531, and 533. To keep it more consistent, articles where additionally removed before common names of other fruit fly species in line 52, 84, 418, 420, 429, 445, 453, 476, 477, and 520.

3. We made the suggested minor grammatical changes by the reviewer in the following lines:

34, 51, 60, 78, 79, 84, 87, 88, 91, 93, 96, 101, 109, 113, 114, 118, 121, 123, 124, 125, 128, 129, 131, 132, 135, 138, 139, 151, 155, 156, 161, 184, 185, 205, 209, 210, 215, 232, 233, 236, 240, 244, 245, 269, 289, 304, 309, 348, 353, 355, 360, 381, 390, 391, 393, 397, 430, 431, 437, 446, 456, 458, 459, 464, 465, 478, 481, 482, 488, 491, 493, 494, 497, 502, 510, 513, 516, 518, 520, 521, 524, and 526.

4. We paraphrased the sentence in line 354, as advised.

5. We don’t think an article in front of third-instar larvae is necessary. Comments were made by the reviewer in this regard in lines’ 36, 41, 103, 166, 167, 221, 257, 296, 297, 347, 360, 361, 363, 371, 402, 419, 435, 499, 501, and 515.

6. We have not changed the spelling of litre (Australian/ British English) to liter (American English) in line 115, because we have used Australian/Bristish English throughout. Same goes for the suggested change from centre to ‘center’ (the latter being American English) in line 198.

7. We have not hyphenated colony forming units as suggested in line 174, because it is typically written without a hyphen and uses the acronym ‘CFUs’ and not ‘C-FUs’, in much of the scientific literature. 

8. Line 178. The authors felt the suggested change was unnecessary and doesn’t improve or alter the readability of the MS. 

9. We decided not to hyphenate surface sterilised pupae as suggested in line 186, as it is not conventional to do this.

10. We have added the species name before isolate names for clarification (Line 299, 302, 348, 349, 360, 378, 481, 500, 501, and 513).

11. We did not make the suggested change in line 470 “Ranging from 5 to 31 days”, as this is implied by the written format and seems redundant to state. 

Response to reviewer 2:

On behalf of the authors, we went through the comments and made changes to clarify content when suggested.

The following comments were addressed, and changes made:

1. We specified the larval stage tested as suggested in line 96.

2. The treatment was specified as suggested in line 129 and 135.

3. To clarify the isolates details, the caption of table 2 was edited (line 345).

4. Isolate details were clarified in line 191, 192 as well as in lines’ 348, 349, as advised.

5. We added information about two field studies. The first study was using M. anisopliae alone and in combination with a spinosad based bait spray against Oriental fruit fly (B. dorsalis) in Kenya, whereas the second study was using a combination of fungal entomopathogens (M. anisopliae or B. bassiana) and entomopathogenic nematodes (S. carpocapsae or H. bacteriophora) against peach fruit fly (B. zonata) and Oriental fruit fly (B. dorsalis) (line 529 and following). These studies suggesting that fungal isolates can be part of an effective IPM approach for field suppression of pestiferous fruit flies. Field efficacy studies of the fungal isolates have been suggested for future studies, however, the manuscript presents a complete body of work, is novel and will add value to field of research.

---

## [Decision Letter · Decision Letter 1]

4 Jan 2024

Metarhizium spp. isolates effective against Queensland fruit fly juvenile life stages in soil.

PONE-D-23-37553R1

Dear Dr. Prince,

We’re pleased to inform you that your manuscript has been judged scientifically suitable for publication and will be formally accepted for publication once it meets all outstanding technical requirements.

Kind regards,

Muthugounder Subramanian Shivakumar, Ph.D.

Academic Editor

PLOS ONE

Additional Editor Comments (optional):

the manuscript can be accepted

Reviewers' comments:

Reviewer's Responses to Questions

**Comments to the Author**

1. If the authors have adequately addressed your comments raised in a previous round of review and you feel that this manuscript is now acceptable for publication, you may indicate that here to bypass the “Comments to the Author” section, enter your conflict of interest statement in the “Confidential to Editor” section, and submit your "Accept" recommendation.

Reviewer #1: All comments have been addressed

2. Is the manuscript technically sound, and do the data support the conclusions?

Reviewer #1: Yes

3. Has the statistical analysis been performed appropriately and rigorously? 

Reviewer #1: Yes

4. Have the authors made all data underlying the findings in their manuscript fully available?

Reviewer #1: Yes

5. Is the manuscript presented in an intelligible fashion and written in standard English?

Reviewer #1: Yes

6. Review Comments to the Author

Reviewer #1: When I reviewed the PDF, I noticed that the authors had made all the necessary corrections. The article can be printed with this version.

My decision is: Accept

7. PLOS authors have the option to publish the peer review history of their article (what does this mean?). If published, this will include your full peer review and any attached files.

Reviewer #1: No

---

## [Editor Report · Acceptance letter]

9 Jan 2024

PONE-D-23-37553R1 

PLOS ONE

Dear Dr. Prince, 

I'm pleased to inform you that your manuscript has been deemed suitable for publication in PLOS ONE. Congratulations! Your manuscript is now being handed over to our production team.

Kind regards, 

on behalf of

Dr. Muthugounder Subramanian Shivakumar 

Academic Editor

PLOS ONE